# Single Domain Antibody-Mediated Blockade of Programmed Death-Ligand 1 on Dendritic Cells Enhances CD8 T-cell Activation and Cytokine Production

**DOI:** 10.3390/vaccines7030085

**Published:** 2019-08-07

**Authors:** Katrijn Broos, Quentin Lecocq, Brenda De Keersmaecker, Geert Raes, Jurgen Corthals, Eva Lion, Kris Thielemans, Nick Devoogdt, Marleen Keyaerts, Karine Breckpot

**Affiliations:** 1Laboratory of Molecular and Cellular Therapy, Department of Biomedical Sciences, Vrije Universiteit Brussel (VUB), 1090 Brussels, Belgium; 2Unit of Cellular and Molecular Immunology, Vrije Universiteit Brussel (VUB), 1090 Brussels, Belgium; 3Myeloid Cell Immunology Lab, VIB Center for Inflammation Research, 1090 Brussels, Belgium; 4Laboratory of Experimental Hematology, Vaccine and Infectious Disease Institute, Faculty of Medicine and Health Sciences, University of Antwerp, 2650 Antwerp, Belgium; 5Center for Cell Therapy and Regenerative Medicine, University Hospital Antwerp, 2650 Antwerp, Belgium; 6In Vivo Cellular and Molecular Imaging Laboratory, VUB, 1090 Brussels, Belgium; 7Nuclear Medicine Department, UZ Brussel, 1090 Brussels, Belgium

**Keywords:** human, PD-L1, dendritic cell, vaccination, single domain antibody, nanobody, cancer, immunotherapy

## Abstract

Dendritic cell [DC] vaccines can induce durable clinical responses, at least in a fraction of previously treated, late stage cancer patients. Several preclinical studies suggest that shielding programmed death-ligand 1 [PD-L1] on the DC surface may be an attractive strategy to extend such clinical benefits to a larger patient population. In this study, we evaluated the use of single domain antibody [sdAb] K2, a high affinity, antagonistic, PD-L1 specific sdAb, for its ability to enhance DC mediated T-cell activation and benchmarked it against the use of the monoclonal antibodies [mAbs], MIH1, 29E.2A3 and avelumab. Similar to mAbs, sdAb K2 enhanced antigen-specific T-cell receptor signaling in PD-1 positive (PD-1^pos^) reporter cells activated by DCs. We further showed that the activation and function of antigen-specific CD8 positive (CD8^pos^) T cells, activated by DCs, was enhanced by inclusion of sdAb K2, but not mAbs. The failure of mAbs to enhance T-cell activation might be explained by their low efficacy to bind PD-L1 on DCs when compared to binding of PD-L1 on non-immune cells, whereas sdAb K2 shows high binding to PD-L1 on immune as well as non-immune cells. These data provide a rationale for the inclusion of sdAb K2 in DC-based immunotherapy strategies.

## 1. Introduction

Dendritic cell [DC] vaccination is extensively studied as a strategy to activate cancer-specific cytotoxic T lymphocytes [CTLs]. To induce potent anti-tumor CTLs three requirements need to be fulfilled. First, the peptide/MHC-I complex on the surface of DCs must be correctly recognized by the T-cell receptor [TCR] expressed on CD8^pos^ T cells. Second, co-stimulatory molecules, like CD80 and CD86, expressed on DCs, need to bind with their receptors, like CD28, expressed on CD8^pos^ T cells. Finally, a third signal is provided by DCs in the form of cytokine secretion. Only when those requirements are fulfilled, the DCs are able to activate T cells capable to attack tumor cells [1].

DCs also express inhibitory molecules, such as programmed death ligand 1 [PD-L1], which binds to its receptor programmed death-1 [PD-1] on activated CTLs, and acts as a brake on T-cell activation [2]. Interaction of PD-L1 with PD-1 during antigen presentation results in TCR down-modulation, preventing T-cell hyperactivation [3,4,5]. However, in the case of cancer vaccination, hyperactivation of T cells is potentially beneficial [3,6,7,8,9].

Several strategies have been successfully employed to interfere with PD-L1/PD-1 interactions during antigen presentation by DCs to CD8^pos^ T cells. These include silencing of PD-L1 [3,10], use of soluble PD-1 or PD-L1 [11,12] and use of antibodies. We recently reported on the development of a camelid single domain antibody [sdAb] that blocks, like the monoclonal antibody [mAb] avelumab, the interaction between PD-1 and PD-L1. This sdAb is referred to as sdAb K2. We showed that sdAb K2 has high potential for non-invasive, radionuclide-based imaging of PD-L1 expressed on tumor cells. We moreover established that sdAb K2 blocks the interaction between PD-1 and PD-L1 at the protein level, and that this blocking ability facilitates killing of tumor cells by cytolytic immune cells present in peripheral blood mononuclear cells [PBMCs] [13]. As sdAbs are versatile antigen binding moieties, we now studied whether sdAb K2 can be used to enhance the activation of tumor antigen-specific CD8^pos^ T cells by monocyte-derived DCs [moDCs].

## 2. Materials and Methods

### 2.1. Reagents

A Melan-A/HLA-A2 dextramer [ELAGIGILTV; Immudex; Copenhagen; Denmark] conjugated to phycoerythrin [PE] was used to detect Melan-A-specific T cells in flow cytometry. A gp100/HLA-A2 dextramer [YLEPGPVTV, Immudex, Copenhagen, Denmark] conjugated to PE was used as a control.

The gp100_280-288_ peptide [YLEPGPVTA, Eurogentec, Cologne, Germany] was used to pulse antigen-presenting cells in the 2D3 assay.

The following blocking anti-PD-L1 mAbs were used in the functional assays: The IgG1 mAbs MIH1 [eBioscience, Brussels, Belgium] and avelumab [Bavencio^®^, Merck KGaA, Overijse, Belgium], and the IgG2b mAb 29E.2A3 [Bioxcell, West Lebanon, NH, USA]. The isotype matched control mAbs P3.6.2.8.1 [IgG1, eBioscience, Brussels, Belgium], MPC 11 [IgG2b, Bioxcell, West Lebanon, NH, USA] or MOPC21 [IgG1, Bioxcell, West Lebanon, NH, USA] were used as controls. The human PD-L1-specific sdAb K2 was previously described [13]. A sdAb specific for the 5T2MM paraprotein, sdAb R3B23, was used as a control [14]. sdAbs were produced and purified as described [15].

Anti-HIS mAbs [Bio-Rad, Temse, Belgium, AD1.1.10] and PE-conjugated anti-mouse IgG antibodies [BD biosciences, Erembodegem, Belgium, A85-1] were used to detect binding of HIS-tagged sdAbs to moDCs and PD-L1 positive (PD-L1^pos^) 293T cells. An anti-mouse IgG2b FITC-conjugated antibody [Southern Biotech, Antwerp, Belgium] was used to detect binding of IgG2b mAb 29E.2A3 on moDCs and PD-L1^pos^ 293T cells.

The following antibodies were used to phenotype cells: An anti-CD3-BV605 [BD biosciences, Erembodegem, Begium, UCHT1], an anti-CD8-APC H7 [BD Biosciences, Erembodegem, Belgium, SK1], an anti-PD-L1-APC [eBioscience, Brussels, Belgium, MIH5], an anti-PD-1-PE [Biolegend, San Diego, CA, USA, EH12.2H7], an anti-HLA-DR-PE-Cy7 [BD Biosciences, Erembodegem, Belgium, G46 6], an anti-CD86-FITC [BD Biosciences, Erembodegem, Belgium, FUN-1], an anti-CD70-PECF594 [BD Biosciences, Erembodegem, Begium, Ki-24], an anti-PD-L2-APC [BD Biosciences, Erembodegem, Belgium, MIH18], an anti-CD80-PerCP-EF710 [eBiosciences, Brussels, Belgium, 2D10.4], an anti-CD11c-AF700 [BD biosciences, Erembodegem, Belgium, clone B ly6], an anti-PD-L1-PE-CF594 [BD Biosciences, Erembodegem, Belgium, clone MIH1], an anti-CD86-BV421 [BD biosciences, Erembodegem, Belgium, clone HB15e], an anti-CD40-APC [Biolegend, San Diego, CA, USA, clone 5C3], an anti-CD80-PerCP-EF710 [eBioscience, Brussels, Belgium, clone 2D10.4] and an anti-HLA-ABC-FITC [BD biosciences, Erembodegem, Belgium, clone G46-2.6]. Isotype matched control antibodies were purchased from BD biosciences.

### 2.2. Cells and Cell Lines

The generation and culture conditions of PD-1 negative (PD-1^neg^) and PD-1^pos^ 2D3 cells were previously described [16]. Culture conditions of 293T cells were described in [15]. Lentiviral vectors encoding the human PD-L1 gene were used to generate PD-L1^pos^ HEK293T cells as described in [15]. Blood from healthy HLA-A*0201^pos^ donors was obtained from the Blood Transfusion Center of the UZ Brussel [Brussels, Belgium]. Isolation of PBMCs, CD14^pos^ monocytes and their differentiation to moDCs, and isolation of CD8^pos^ T cells from the remaining PBMCs was performed as described [17]. This study was approved by the Ethical Committee of the UZ Brussel [2013/198].

### 2.3. mRNA Production and Electroporation

The pGEM-vectors encoding the human gp100 TCRα and TCRβ were kindly provided by Prof. N. Schaft [Universitätsklinikum Erlangen, Erlangen, Germany]. The pGEM-sig-Melan-A-DC-LAMP plasmid encoding the full-length Melan-A antigen containing the optimized immunodominant Melan-A/HLA-A2 epitope linked to the HLA-II targeting sequence of DC-LAMP was described [18]. The production, purification, quantification and quality control of mRNA were performed as described [17].

Human gp100 TCRα and TCRβ mRNA [2.5 µg each/10^6^ cells] was electroporated into PD-1^neg^ and PD-1^pos^ 2D3 cells in 200 µL OptiMEM medium [Life Technologies, Erembodegem, Belgium] in a 4 mm electroporation cuvette [Cell Projects] using a time constant protocol [300 V, 7 ms] and the Gene Pulser Xcell^TM^ device [Biorad, Temse, Belgium]. Electroporation of moDCs with mRNA [2.5 µg/component/10^6^ cells] was performed as described [17].

### 2.4. 2D3 Assay

The 2D3 assay was performed as described [16]. Briefly, PD-1^neg^ and PD-1^pos^ 2D3 cells, electroporated to express the TCR, recognizing the gp100_280–288_ peptide [YLEPGPVTA] restricted to HLA-A2, were plated in triplicate in a 96 well round bottom plate at 10^5^ cells per well in 200 µL Iscove’s modified Dulbecco’s medium [IMDM, Gibco Invitrogen] containing 10% fetal bovine serum [FBS, Biochrom AG]. moDCs were pulsed with 50 µg/mL gp100_280–288_ peptide and added to the cultures at effector stimulator ratios of 10:1 in 100 µL medium. Co-cultures were performed for 24 h at 37 °C, 5% CO_2_ in the presence of 1 µg/200 µL blocking anti-PD-L1 mAbs or sdAb K2, or isotype matched control mAbs or sdAb R3B23. Signaling of the TCR in 2D3 cells was measured by flow cytometry as percentage enhanced green fluorescence protein positive [eGFP^pos^] cells within CD8^pos^ 2D3 cells.

### 2.5. Stimulation of CD8^pos^ Melan-A-Specific T Cells by DCs

CD8^pos^ T cells were plated in triplicate in a 96 well round bottom plate at 10^5^ cells per well in 100 µL IMDM containing 1% heat-inactivated human AB serum [Innovative Research, Novi, MI, USA], 100 U/mL penicillin, 100 µg/mL streptomycin, 2 mM L-Glutamine and non-essential amino acids [Sigma Aldrich]. moDCs were electroporated with Melan-A mRNA to generate so called DC-MEL [19]. DC-MEL were added to the T cells at an effector:stimulator ratio of 10:1 in 100 µL culture medium. Co-cultures with DC-MEL were performed for 7 days at 37 °C, 5% CO_2_ in the presence of 10 µg/200 µL blocking anti-PD-L1 mAbs or sdAb K2, or isotype matched control mAbs or sdAb R3B23. T cells were restimulated on day 7. Analysis of the activation of Melan-A-specific T cells was performed 7 days after the last stimulation. The number of viable, Melan-A/HLA-A2 dextramer^pos^ CD8^pos^ T cells was determined by flow cytometry. The production of interferon γ [IFN-γ] was evaluated with ELISA [Thermo Scientific, Brussels, Belgium] according to the manufacturer’s instructions. The production of interleukin [IL]-2, IL-4 and tumor necrosis factor-α [TNF-α] was determined using BioPlex Pro Human Cytokines [Biorad, Temse, Belgium] according to the manufacturer’s instructions.

### 2.6. Proliferation Assay

PBMCs depleted from CD14^pos^ cells from healthy donor were labeled with 0.5 µM Cell Trace Violet (Invitrogen). These cells (10^5^) were co-cultured for 6 days with or without DC-MEL (4.8 µg Melan-A mRNA) at a effector:stimulator ratio of 10:1 in 200 µL IMDM containing 1% heat-inactivated human AB serum, PS, L-Glu and non-essential amino acids. T-cell proliferation was measured in flow cytometry as the dilution of the CellTrace Violet dye in the CD8^pos^ T-cell population. Proliferation observed in cultures without DC-MEL was considered as background.

### 2.7. Evaluation of DC Maturation in Response to Endotoxins Present in sdAb Preparations

To evaluate the effect of any endotoxins in the sdAb solutions, we incubated moDCs for 24 h with 10 µg sdAb K2 or sdAb R3B23 at 37 °C and 5% CO_2_. Untreated moDCs and moDCs treated with 1 ng/mL lipopolysaccharide [LPS] served as negative and positive controls, respectively. Up-regulation of maturation markers was evaluated in flow cytometry.

### 2.8. Flow Cytometry

The procedure for staining of surface markers was previously described [20]. All cells were acquired on the LSRFortessa flow cytometer [BD Biosciences, Erembodegem, Belgium] and data were analyzed with FACSDiva [BD Biosciences] or FlowJo [Tristar Inc.] software.

### 2.9. Statistical Analysis

Results are expressed as mean ± standard error of the mean [SEM]. A paired student t-test was carried out to compare data sets. Sample sizes and number of independent experiments are indicated in the figure legends. The number of asterisks in the figures indicates the statistical significance as follows: * *p* < 0.05; ** *p* < 0.01 and *** *p* < 0.001.

## 3. Results

### 3.1. Inhibition of TCR Signaling in PD-1^pos^ 2D3 Cells Activated with PD-L1^pos^ DCs is Alleviated by sdAb K2

To evaluate whether sdAb K2 can be used as a therapeutic agent in combination with DC-vaccination, we first performed a functional assay using PD-1^pos^ and PD-1^neg^ 2D3 cells. The latter are derived from the established Jurkat T-cell line and are characterized by CD8 expression, lack of endogenous TCR expression and expression of eGFP under the control of the nuclear factor of activated T cells [NFAT] promoter. We previously showed that TCR modified PD-1^pos^ and PD-1^neg^ 2D3 cells can be used to validate the blocking capacity of PD-1 and PD-L1 blocking mAbs [16].

PD-1^pos^ and PD-1^neg^ 2D3 cells, electroporated with mRNA encoding the TCR recognizing gp100_280-288_ in the context of HLA-A2 (Figure 1a,b), were co-cultured with HLA-A2^pos^ PD-L1^pos^ moDCs pulsed with gp100_280-288_ peptide (Figure 1c). The expression of eGFP was determined 24 h later as a measure of TCR signaling. We observed that the expression of eGFP by 2D3 cells was inhibited upon PD-1/PD-L1 interaction (Figure 1d). This inhibition could be alleviated through addition of the anti-PD-L1 mAb MIH1 [IgG1] or sdAb K2, but not through addition of an isotype matched control mAb or the control sdAb R3B23 (Figure 1d). Comparable results as with mAb MIH1 and sdAb K2 were obtained with the IgG1 mAb avelumab. To ensure that the activation of 2D3 cells in the context of moDC stimulation and sdAb K2 mediated PD-1/PD-L1 blockade was not due to maturation of the moDCs as a result of endotoxins present in the sdAb preparations, we compared the phenotype of moDCs that were untreated or matured with LPS to the phenotype of moDCs treated with sdAb K2 or sdAb R3B23. Up-regulation of maturation associated phenotypic markers like CD40, CD80 and the antigen presenting molecule HLA-I were only observed when moDCs were treated with LPS (Figure 1e). These results indicate that the increase in TCR signaling in PD-1^pos^ 2D3 cells during antigen presentation by PD-L1^pos^ moDCs in the presence of sdAb K2 is most likely due to inhibition of the PD-1/PD-L1 interaction and not due to an increase in HLA-I expression, therefore antigen presentation.

In conclusion, sdAb K2 potently enhances TCR-signaling during antigen presentation by moDCs, as shown by the NFAT-mediated up-regulation of eGFP in PD-1^pos^ 2D3 cells.

### 3.2. Activation of Melan-A-Specific CD8^pos^ T Cells by DC-MEL Is Significantly Enhanced by sdAb K2-Mediated PD-1/PD-L1 Blockade

To address whether PD-L1 blockade enhances the activation of antigen-specific CD8^pos^ T cells by DC-MEL, i.e., moDCs electroporated with mRNA encoding Melan-A, we performed T-cell stimulation experiments. Herein DC-MEL were co-cultured with CD8^pos^ T cells at a 1 to 10 ratio. Two rounds of stimulation were performed in the presence of the blocking anti-PD-L1 mAbs 29E.2A3 or avelumab, isotype control mAbs, sdAb K2 or sdAb R3B23 (control) to obtain sufficient Melan-A-specific T cells for analysis. Of note, avelumab was chosen for comparison to sdAb K2 as we previously showed that avelumab and sdAb K2 compete for binding to PD-L1 [13].

First, we phenotyped the DC-MEL using flow cytometry, analyzing the expression of the co-inhibitory molecule PD-L1 as well as the co-stimulatory molecules CD70, CD80 and CD86, and the antigen presenting molecules HLA-A2 and HLA-DR. We observed expression of all evaluated surface markers on DC-MEL with the exception of the co-stimulatory molecule CD70 (Figure 2a and Figure 3a). Furthermore, we used flow cytometry to analyze the expression of PD-1 on the CD8^pos^ T cells before they were co-cultured with DC-MEL, showing low expression of PD-1 (Figure 2b and Figure 3b). Detection of Melan-A-specific CD8^pos^ T-cell activation upon co-culture with DC-MEL in the presence of blocking anti-PD-L1 mAbs 29E.2A3 or avelumab, isotype control mAbs, sdAb K2 or sdAb R3B23 (control) was performed using flow cytometry. Together with the total number of cells this allowed us to determine the number of Melan-A-specific T cells that were expanded in these co-cultures. We showed that compared to respective control compounds, the presence of sdAb K2 but not the anti-PD-L1 mAb 29E.2A3 or avelumab resulted in a significantly higher number of Melan-A-specific T cells (Figure 3c). We further analyzed the capacity of the activated Melan-A-specific CD8^pos^ T cells to produce multiple cytokines and to proliferate using multiplex cytokine analysis and evaluation of Cell Trace Violet dilution in flow cytometry, respectively. We observed that Melan-A-specific CD8^pos^ T cells stimulated in the presence of sdAb K2 secreted more IL-2, IL-10, IFN-γ and TNF-α when compared to CD8^pos^ T cells stimulated in the presence of sdAb R3B23. Of these cytokines IL-2 and IL-10 were produced at pg-levels, while IFN-γ and TNF-α were produced at ng-levels (Figure 3d). Moreover, these CD8^pos^ T cells showed a higher proliferative capacity as well (Figure 3e). These data suggest that sdAb K2, however not mAb 29E.2A3 or avelumab, can block the interaction between PD-L1 and PD-1 during antigen presentation by DC-MEL to CD8^pos^ T cells, as such resulting in enhanced activation of functional antigen-specific CD8^pos^ T cells.

It was unexpected that the mAbs 29E.2A3 and avelumab were unable to enhance the activation of Melan-A-specific CD8^pos^ T cells by DC-MEL, as both have been described as blocking mAbs to enhance activation of antigen-specific CD8^pos^ T cells by PBMCs [21,22]. Moreover, avelumab has proven its efficiency in several clinical trials [23]. In search for an explanation for this lack of effect, we evaluated mAbs 29E.2A3 and avelumab as reagents to detect and bind human PD-L1 on moDCs. Staining of moDCs with mAb MIH1 and sdAb K2 were performed for comparison. We observed that mAb MIH1 was most efficient as a reagent to detect PD-L1 on moDCs followed by sdAb K2, avelumab and mAb 29E.2A3 (Figure 4a). In fact, in flow cytometry mAb 29E.2A3 was not proficient as a reagent to stain PD-L1. In contrast, efficient staining of PD-L1 on PD-L1^pos^ 293T cells was observed (Figure 4b), suggesting a different efficiency of mAb 29E.2A3 and avelumab to bind to PD-L1 on immune cells versus non-immune cells. Such differences in sensitivity of reagents to discriminate PD-L1 on immune cells versus tumor cells in immunohistochemistry was previously described [24].

In conclusion, these results suggest that the low efficacy of 29E.2A3 and avelumab to bind PD-L1 on moDCs when compared to sdAb K2 may impact on the efficacy of mAb 29E.2A3 and avelumab to enhance DC-mediated T-cell activation. Furthermore, the results generated with sdAb K2 in the context of TriMixDC-MEL and DC-MEL mediated CD8^pos^ T-cell activation suggest that in the absence of strong co-stimulatory signals, PD-L1 is a major determinant of T-cell activation. Finally, the data generated with DC-MEL and sdAb K2 provides a rationale for the inclusion of sdAb K2 in DC-based immunotherapy strategies.

## 4. Discussion

The inhibitory function of PD-1/PD-L1 interaction during antigen presentation by DCs to T cells is generally recognized, pinpointing this inhibitory pathway as an attractive therapeutic target to enhance the potency of DC-vaccines. Several strategies have been successfully employed in preclinical studies to interfere with PD-1/PD-L1 interactions during antigen presentation by DCs to CD8^pos^ T cells. In particular the use of mAbs in combination with DC-vaccination has found its way to the clinic, as evidenced by a number of clinical trials in a range of malignancies [25]. However, the immune synapse clears and even excludes molecules above a certain size, including mAbs [26]. Therefore, the use of small sized, blocking PD-1/PD-L1 agents might be more advantageous. Recently, we described a human PD-L1 specific sdAb, sdAb K2, obtained after alpaca immunizations and biopannings, as a PD-1/PD-L1 neutralizing moiety with high target specificity and affinity [13]. Its small size [≈15 kDa, 10 times smaller than a mAb] was shown to penetrate much more efficiently within cell-cell interfaces like immune synapses [26], making it an interesting candidate for implementation in combination therapy with DCs.

We showed that sdAb K2 could shift the balance between stimulatory and inhibitory signals during the early stage of T-cell activation when using DCs with a low stimulatory profile. Although several mAbs targeting PD-L1 have shown clinical efficacy, we were unable to show any benefit of adding mAbs to the in vitro DC-MEL CD8^pos^ T-cell co-cultures [23]. This is in contrast to other studies reporting on the use of avelumab and/or mAb 29E.2A3 to enhance the activation of human T-cell populations of healthy donors [21,22]. Several reasons can explain this discrepancy. Grenga et al. [21] studied interaction between PBMCs and CD8^pos^ T cells, rather than moDCs and CD8^pos^ T cells, showing that in this setting, activation of virus-specific CD8^pos^ T cells was most pronounced in the presence of avelumab when compared to mAb 29E.2A3. The use of viral peptides as antigens is a major difference, as in this case most likely memory CD8^pos^ T cells are activated rather than naïve T cells. They claim that avelumab is able to shift Th2 cells to Th1 cells, demonstrated by an increase in IFN-γ concentration. The maximum amount of IFN-γ and the hereby increase in IFN-γ (±2 ng/mL) was comparable with the increases in IFN-γ after adding sdAb K2 to the DC-MEL co-culture (±3 ng/mL). On the other hand, Brown et al. [22] used moDCs as stimulator cells, however, performed allogeneic mixed lymphocyte reactions, evaluating CD4^pos^ T cell activation in the presence of the mAb 29E.2A3. In this setting the presence of allogeneic HLA-antigens may serve as a danger signal, also inducing overall T-cell activation, including memory T-cell activation [27]. Re-activation of antigen experienced effector memory T cells was suggested to be the driver of the efficacy of PD-L1/PD1 blockade in human cancer therapy, while activation of CD8^pos^ effector T cells was not reported [28]. Nonetheless, here as well, the level of IFN-γ concentration observed after treatment with mAb 29E.2A3 (5 ng/mL; 40 μg/mL mAb 29E.2A3) was at the same level as observed in the DC-MEL co-cultures after treatment with 50 μg/mL sdAb K2 (± 3 ng/mL). Moreover, the concentration of blocking mAbs used, differs from the above described studies. Both Grenga et al. and Brown et al. observed an effect when using 20 μg/mL of anti-PD-L1 mAbs, whereas we could not even observe an effect when using 50 μg/mL. Moreover, in the in vitro cell-based assay, DC-MEL were exclusively co-cultured with CD8^pos^ T cells, therefore not taking into account the role of PD-L1 blockade on other immune cells such as myeloid-derived suppressor cells, macrophages, etc. It was previously shown that avelumab exerts its effect via antibody-dependent cellular cytotoxicity [ADCC] via three types of Fc receptors, FcγRI [CD64], FcγRII [CD32], and FcγRIIIA [CD16]. FcγRIIIA [CD16] is pinpointed as the main player in ADCC as it is expressed predominantly by natural killer cells, which were not represented in the in vitro cell culture assays [29,30]. The source of stimulator cells might also contribute to the difference in experimental outcome. We observed that mAb 29E.2A3 was unable to detect PD-L1 in flow cytometry on the moDCs we generated, and that detection of PD-L1 with avelumab was less evident than with mAb MIH1 and sdAb K2. Staining of PD-L1 expressed on 293T cells precludes that this observation is a technical artefact, as avelumab, mAbs 29E.2A3 and MIH1 as well as sdAb K2, were able to stain PD-L1 on these cells. The reason for this different sensitivity to PD-L1 expressed on moDCs versus 293T cells is at present unclear, however, might explain as to why de novo activation of antigen-specific CD8^pos^ T cells was not observed in the presence of these mAbs in our study. It is conceivable that our moDCs differ from the moDCs used by Brown et al. [22], as different culture conditions were used [e.g., culture medium]. For sure, the moDCs used in this study are different from the PBMCs used by Grenga et al. [21]. Further studies are required to assess binding of different mAbs to different PD-L1^pos^ immune and non-immune cell populations, including DCs. Such studies have already been performed with other antibodies in the context of immunohistochemical detection of PD-L1 in tumor tissue and lymph nodes, showing that different antibodies indeed have different propensities to bind PD-L1 on tumor cells versus immune cells, and sometimes even discriminate between lymphocyte-like cells versus DCs [24]. In our study, sdAb K2, similar to mAb MIH1, did not make the distinction between PD-L1 expressed on moDCs versus 293T cells.

We previously showed that both sdAb K2 and avelumab bind with high affinity to PD-L1 and that binding of K2 hampers subsequent binding of avelumab and vice versa [13]. Studies showed that avelumab binds mainly with its V_H_ domain on the strands of the front β sheet face of the IgV domain of PD-L1, which is different from the epitopes bound by other PD-L1 targeting mAbs, such as durvalumab, atezolizumab and BMS-936559 [31,32]. As such sdAb K2 is a unique small-sized, biological inhibitor when compared to other small molecule inhibitors such as the anti-PD-L1 sdAb KN035 and the non-peptide anti-PD-L1 inhibitors BMS-202 and BMS-8, which show similar binding to PD-L1 as durvalumab [32,33]. We previously showed that sdAb K2 blocks PD-1/PD-L1 interactions at the protein level and tumor cell T-cell level [13]. We now show that sdAb K2 can also block PD-1/PD-L1 interactions at the immunological synapse created when DCs interact with CD8^pos^ T cells. The single domain nature of sdAbs offers interesting perspectives in view of DC-vaccine development. Many protocols are available to deliver tumor antigens and activation stimuli to DCs. Many of these are based on genetic engineering using viral and non-viral vectors [34,35]. While cloning of classical mAbs or mAb fragments offers serious challenges, cloning of sdAbs [i.e., coded by a single 360 bp DNA fragment] is straightforward, therefore can be easily incorporated into existing DC-engineering protocols. Several of these ex vivo DC-engineering strategies have also been used to specifically engineer DCs in situ, even in the tumor environment [36,37,38,39,40]. The targeted delivery of sdAb K2, and its release in the immunological synapse offers attractive safety considerations compared to systemic mAb or sdAb-administration. It will tip the balance from immune inhibitory to stimulatory signals only between antigen and sdAb K2-engineered DCs and cognate T cells, thereby ensuring increased on-target T-cell responses with little to no off-target T-cell activation.

In conclusion, we report on the use of sdAb K2, a versatile PD-L1/PD-1 blocking moiety, to enhance the capacity of DCs to stimulate T-cell activation and cytokine production. Inclusion of sdAb K2 in DC-vaccination protocols may have therapeutic potential in the clinical setting where several technologies to modify DCs for T-cell activation are investigated in the setting of cancer as well as infectious disease.

## 5. Patents

The use of moDCs electroporated with tumor antigen and TriMix mRNA is the topic of a patent application [WO2009/034172] on which K.T. is filed as an inventor. This patent is licensed to eTheRNA Immunotherapies NV. The use of sdAb K2 in onco-immunology is the topic of a patent application [EP18208646.2] on which K.B. (Katrijn Broos), K.B. (Karine Breckpot), N.D., M.K., Q.L. and G.R. are filed as inventors. None of the authors receive any support or remuneration related to this platform.

## Figures and Tables

**Figure 1 vaccines-07-00085-f001:**
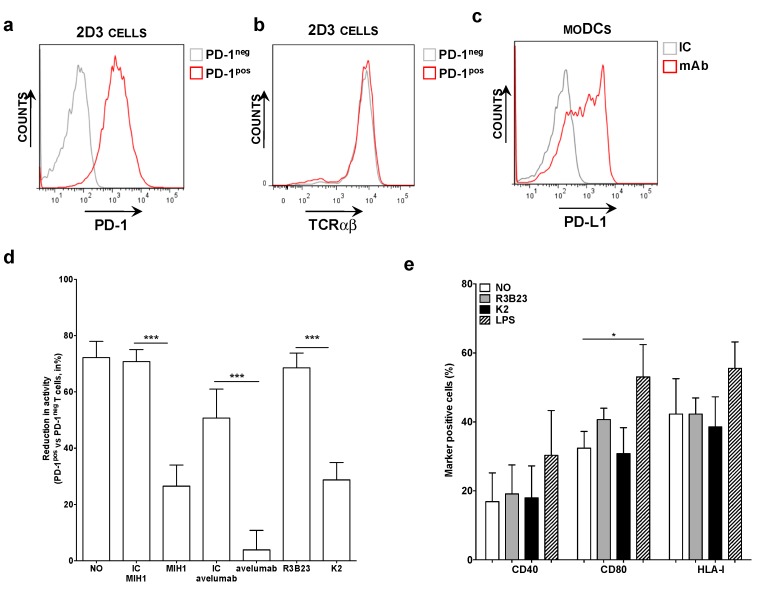
Antigen-specific activation of TCR^pos^ PD-1^pos^ 2D3 cells by PD-L1^pos^ moDCs is enhanced in the presence of blocking anti-PD-L1 mAbs or sdAb K2. (**a**) Histogram showing PD-1^neg^ [grey] and PD-1^pos^ [red] 2D3 cells stained with anti-PD-1 mAbs. These are representative for 6 independent experiments. (**b**) Histogram showing the expression of the TCR recognizing gp100 in the context of HLA-A2 on PD-1^neg^ [grey] and PD-1^pos^ [red] 2D3 cells stained with anti-TCR mAbs. These are representative for 6 independent experiments. (**c**) Representative histogram showing PD-L1 expression on moDCs. In three independent experiments, cells were stained with isotype control [IC, grey] or anti-PD-L1 [red] mAbs [n = 3]. (**d**) Reduction in TCR signaling in PD-1^pos^ TCR^pos^ versus PD-1^neg^ TCR^pos^ 2D3 cells when activated with antigen presenting moDCs, calculated as [1 − (%CD8^pos^ eGFP^pos^ PD-1^pos^ TCR^pos^ 2D3 cells/% CD8^pos^ eGFP^pos^ PD-1^neg^ TCR^pos^ 2D3 cells)] * 100%. The x-axis legend represents co-cultures without addition of mAbs or sdAbs [no], or with addition of isotype control mAbs [IC], the anti-PD-L1 mAb [MIH1 or avelumab], sdAb R3B23 [R3B23] or sdAb K2 [K2]. The graph summarizes the reduction in TCR signaling as mean ± SEM of three independent experiments. (**e**) Graph showing the percentage of moDCs that express CD40, CD80 and HLA-I when untreated [no], treated with sdAb R3B23 [R3B23], sdAb K2 [K2], or LPS. The graph summarizes the percentage surface marker expression as mean ± SEM of four independent experiments. The number of asterisks in the figures indicates the statistical significance as follows: * *p* < 0.05; ** *p* < 0.01 and *** *p* < 0.001.

**Figure 2 vaccines-07-00085-f002:**
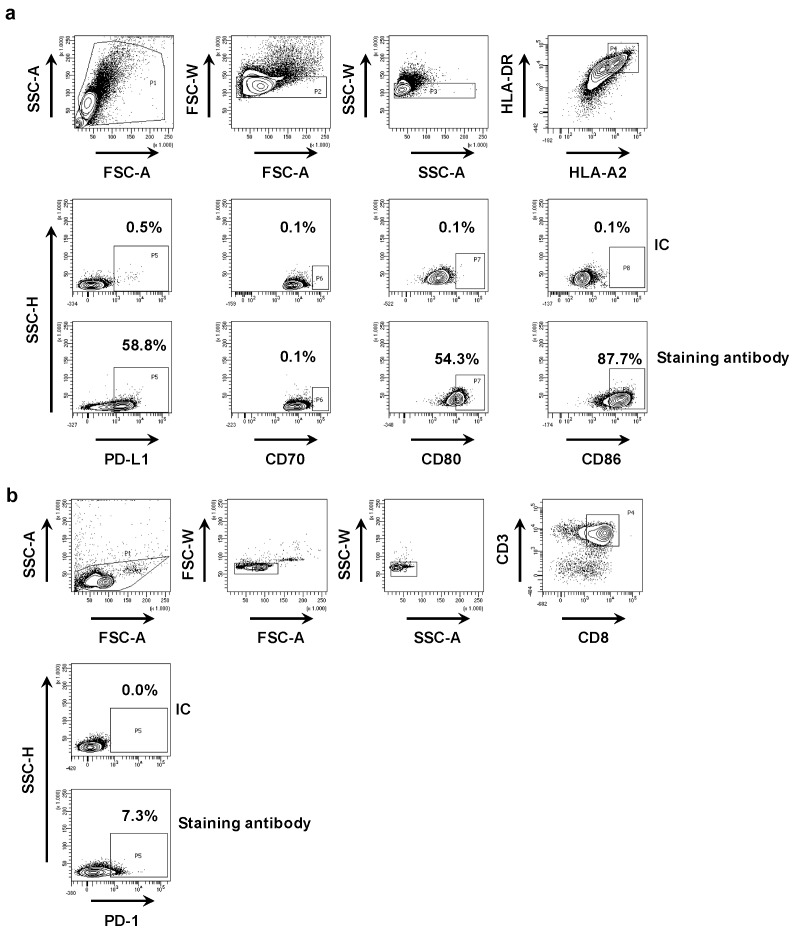
Phenotyping of DC-MEL and CD8^pos^ T cells. (**a**) Representative flow cytometry graphs showing the gating strategy used to phenotype DC-MEL. Viable cells were selected based on SSC-A/FSC-A characteristics. Subsequently single cells were selected based on FSC-A/FSC-W and SSC-A/SSC-W characteristics. After gating of single cells, PD-L1, CD70, CD80 and CD86 expression was evaluated on HLA-DR^pos^ HLA-A2^pos^ cells. To set the gate that defines marker^pos^ cells, we used cells stained with the anti-HLA-DR and anti-HLA-A2 mAbs together with isotype control mAbs [IC] coupled to the same fluorophore as the mAbs specific for PD-L1, CD70, CD80 or CD86 [antibody staining]. (**b**) Representative flow cytometry graphs showing the gating strategy used to detect PD-1 expression on CD8^pos^ T cells. Viable cells were selected based on SSC-A/FSC-A characteristics. Subsequently single cells were selected based on FSC-A/FSC-W and SSC-A/SSC-W characteristics. After gating of single cells, PD-1 expression was evaluated within the population of CD3^pos^ CD8^pos^ T cells. To set the gate that defines PD-1^pos^ cells, we used cells stained with the anti-CD3 and anti-CD8 mAbs together with isotype control mAbs [IC] coupled to the same fluorophore as the anti-PD-1 mAb [antibody staining].

**Figure 3 vaccines-07-00085-f003:**
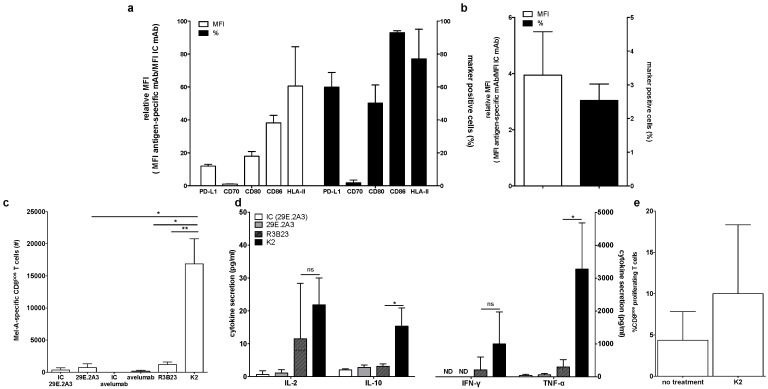
Activation of Melan-A-specific CD8^pos^ T cells by DC-MEL is enhanced in the presence of sdAb K2. (**a**) Graph showing the percentage of marker positive cells as well as the relative MFI (MFI antigen-specific mAb/MFI IC mAb) for the expression of PD-L1, CD70, CD80, CD86, and HLA-II on DC-MEL. The graph summarizes the results of three independent experiments as mean ± SEM. (**b**) Graph showing the percentage of PD-1^pos^ cells as well as the relative MFI (MFI antigen-specific mAb/MFI IC mAb) for the expression of PD-1 on T cells. The graph summarizes the results of three independent experiments as mean ± SEM (left). (**c**) Graph showing the number Melan-A-specific T cells after co-culture with DC-MEL in the presence of IC mAbs [IC], anti-PD-L1 mAbs [29E.2A3 and avelumab], sdAb R3B23 [R3B23] or sdAb K2 [K2]. The graph summarizes the results of three independent experiments as mean ± SEM. (**d**) Graph showing the production of IL-2, IL-10, IFN-γ and TNF-α by Melan-A-specific CD8^pos^ T cells co-cultured with DC-MEL in the presence of sdAb K2 compared to sdAb R3B23. The graph summarizes the results of three independent experiments as mean ± SEM. (**e**) Graph showing the percentage of CD8^pos^ proliferating T cells co-cultured with DC-MEL in the presence of sdAb K2. The graph summarizes the results of two independent experiments as mean ± SEM. The number of asterisks in the figures indicates the statistical significance as follows: * *p* < 0.05; ** *p* < 0.01 and *** *p* < 0.001.

**Figure 4 vaccines-07-00085-f004:**
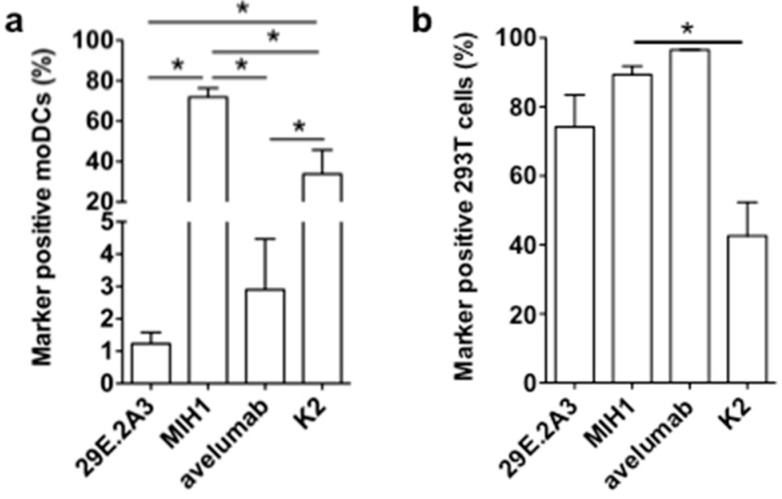
Binding of anti-PD-L1 mAbs and sdAb K2 on PD-L1^pos^ moDCs versus 293T cells. Graph summarizing the percentage PD-L1^pos^ moDCs (**a**) or 293T cells (**b**) detected in flow cytometry upon staining with the mAbs 29E.2A3, MIH1 or avelumab, or sdAb K2. Cells stained with isotype matched control mAbs or sdAb R3B23 served as a control. The graphs summarize the results of three independent experiments. The number of asterisks in the figures indicates the statistical significance as follows: * *p* < 0.05; ** *p* < 0.01 and *** *p* < 0.001.

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
