# Peer review of "Single Domain Antibody-Mediated Blockade of Programmed Death-Ligand 1 on Dendritic Cells Enhances CD8 T-cell Activation and Cytokine Production"

_vaccines, 2019, doi:10.3390/vaccines7030085_

Round 1

Reviewer 1 Report

This papers documents the potential of a camelid anti-PDL1 single-domain Ab to boost CD8+ T-cell responses induced by DCs. It also touches upon the differential binding affinity of anti-PDL1 Abs for different cellular targets. The data is intriguing, but several questions remain answered.

Below are my comments:

47-50: some redundancy in the explanation, can be shortened

Fig.1,2,3 : what are the replicates (“n=6”, “n=3”, …) exactly referring to? Number of different buffy coat donors? Well replicates with cells from same donor? Please be more clear in the legends. Given the donor-to-donor variability, 3 buffy coats is really a bare minimum of independent experiments.

Fig. 1C. The vertical axis label on is misleading. What is represented is not the “reduction in TCR signalling”, otherwise a bar height at e.g. 0.8 would mean a 80% reduction. A label such as “relative TCR activation”(with the emphasis on relative as it is a ratio) would be more appropriate.

Fig 2,3: consider combining into one figure. 2.B seems a little bit out of place with that respect

Fig 2.A, 3.A: Representative histograms are OK, but given the fact there are replicates (donors?) the authors should display surface expression levels of PD-L1, CD70, CD86 as relative MFI (mean +/- SEM). This allows a reasonable comparison between TriMix-DC-Mels and DC-Mels. Also, FMO should be used as background, not isotype control staining.

Fig. 2.C, 3.B: How do the authors explain that tetramer+ CD8 T-cells expansion with anti-PD-L1-treated DC-Mels seem to be larger, or is at least as robust as with anti-PD-L1-treated TriMix DC-Mels?

Fig. 2.D, 3.C. What is fold change in IFN-g, i.e. what is the control condition? What are the absolute levels of IFN-g in all these conditions? Why not use intracellular flow-cytometry to quantify IFN-g production within the tetramer+ T-cells?

Overall, the T-cell activation endpoints reported in this paper are somewhat minimalistic. A more comprehensive reporting of T-cell status in the different stimulation conditions should include proliferation (eg CFSE dilution or equivalent), polyfunctionality (IFNg/TNFa/IL2 expression), markers reflecting acquisition of effector phenotype

258-260: Please provide information on the “PD-L1-pos” 293T cells in the Materials section. If this a a stably transduced HEK 293T, a more clinically relevant model to compare Abs affinity to PD-L1 would be an IFN-g-pretreated carcinoma cell line.

General comments:

The differential binding affinity for a given Ab for PD-L1 expressed on tumor cells vs myeloid cells remains somewhat of a mystery. It is frustrating that this paper, with its quite extensive comparisons between different Abs and different target cells, does not provide at least some experimental data to explain this. After all, PD-L1 is PD-L1, or are there splice variants differentiating cancer cells and immune cells? Or different PD-L1 isoforms when induced by oncogenic pathways or hypoxia versus induction by IFN-g? Are there differences between cell types with respect to cell membrane PD-L1 turnover or recycling?

Why is avelumab used as a reference clinical-grade anti-PDL1? It is one of the least commonly used anti-PDL1 compounds in the clinic compared to atezolizumab or durvalumab

Author Response

To: The Editor of Vaccines,

Concerning: Revision of the manuscript entitled “Single domain antibody-mediated blockade of programmed death-ligand 1 on dendritic cells enhances CD8 T-cell activation and cytokine production”, reference 540420.

Dear Editor,

We would like to thank the editor for considering our manuscript for publication in Vaccines. We furthermore thank all reviewers for their valuable remarks. We believe that by taking the reviewers’ comments into account, we were able to strengthen the manuscript. Therefore, we hope that the manuscript in its present form fits all requirements for publication in Vaccines.

Enclosed, we provide our feedback to the reviewers’ questions in a point-by-point reply.

Sincerely,

Katrijn Broos

Reviewer 2 Report

Broos et al present a well written study, demonstrating the in vitro ability of the single-domain antibody K2 to enhance CD8 T cell activation and cytokine production in co-cultures with DCs. The data are clearly presented, and the caveats of the study are clearly discussed. In general the conclusions are supported by the data but in specific cases additional controls/information may be required.

Major Comments:

-        Figure 1C – Data presentation here is rather confusing. The axis label is “reduction in TCR signalling”, so a lower value indicates a block in the inhibition of signalling? It may be clearer to show the raw data (PD-1neg vs PD-1pos), or to rethink the axis label.

-        Figure 2+3 – The authors draw comparisons between the TriMixDC-MEL and standard DC-MEL. In order to make this comparison, the authors should comment on if the donors for the DCs are the same between the two experiments, if the experiments were performed in parallel and also show data from Figure 2 using the same restimulation as used in Figure 3.

-        Avelumab is a clinically used antibody – if the authors would like make comparisons between K2 and avelumab this data does need to be shown (line 165, line 250).

-        Figure 4 – This binding data is quite striking. Do the TriMixDC-Mel show the same binding pattern as moDC? A negative control (i.e. a PD-L1 negative immune cell/line) would also be helpful in accounting for any non-specific binding that could explain this result.

Minor Comments:

-        The authors state n=2 etc. but do not seem to be explicit about whether these are technical replicates or individual healthy PBMC donors.

Author Response

(The authors gave the same response as above.)

Reviewer 3 Report

The study by Broos et al investigates the use of the PD-L1 specific single domain antibody K2 in enhancing DC-mediated T-cell activation in vitro. The authors compared the ability of K2 to enhance T-cell activation with mAbs MIH1, 29E.2A3 and avelumab.

The study is well conducted with good controls and the manuscript is well written with clear results. There were a few points where clarification could help/ improvements could be made:

1.  Line 49: “However, in the case of cancer vaccination, hyperactivation of T-cells is potentially beneficial.”: It would be useful for readers from outside of the cancer vaccine field to have some references to support this.

2. Methods: A gating strategy for the cell phenotyping should be included in the supplementary information.

3.  Line 165: Given that avelumab has been used clinically to block PD-1-PD-L1 interactions in vivo and the data comparing avelumab with K2 is available, I think it would be beneficial to include this data in the manuscript.

4. Line 174: The authors have demonstrated that the increase in TCR signalling in PD-1pos 2D3 cells during antigen presentation by PD-L1pos moDCs in the presence of sdAb K2 was not due to increased antigen presentation caused by an increase in HLA-I expression. However, was the level of TCR expression comparable on PD-1pos and PD-1neg 2D3 cells? It would be useful to show or comment on this.

5. Figure 1: Figure 1c shows the mean and SEM of 3 experiments. Figure 1d only shows the mean and SEM of 2 experiments. Given the small “n” and the size of the error bars for the % of CD40+ and CD80+ cells, it is difficult to say that these markers were increased in the LPS treated condition but not in the others. It would be more believable if there was data available from more replicates. Additionally, as these are presented as bar charts, not all the data can be seen and this may be misleading.

6. Figure 2d and 3c: Why has fold-increase in IFNg production been used instead of IFNg concentration as in previous studies? Was there an issue with different background levels of IFNg production observed with the isotype control mAb vs sdAb R3B23?

7. Line 210 and lines 272-276: The authors state that the results in Figure 2 suggest that the co-inhibitory signal provided by PD-L1 is a lesser determinant in the degree of T-cell activation when co-stimulatory signals are abundantly provided (as by TriMixDC-MEL). However, as well as lower expression of CD70, DC-MEL also appear to have a lower expression of PD-L1 than TriMixDC-MEL (Figures 2a and 3a). Therefore, could it be possible that this lower PD-L1 expression is also contributing to the results here?

8. Figure 2d: The sample size is only 2, although it was 5 for Figure 2c, which shows antigen-specific T-cell numbers after co-culture with TriMix\DC-MEL in the presence of mAbs or sdAb. Why was IFNg only measured as an output for some of these assays?

9. Line 236: As the authors state that previous results have shown that expansion of CD8pos T-cells in cultures with PBMCs was more pronounced when using mAbs with an IgG1 isotype compared with IgG2b isotype mAb 29E.2A3, it would be useful to include the data for avelumab and MIH1 in Figure 3 (or in a supplementary figure) so the comparison can be seen.

10. Line 235: Do the authors have any data on the percentage of CD8pos T-cells producing IFNg? There is a large expansion of antigen-specific CD8pos T-cells in the presence of sdAb, which is very clear. However, it is unclear if many of these cells are stimulated to produce IFNg or whether a small percentage of these cells are producing lots of IFNg.

11. Figure 3c: This is a very clear difference. Is there statistical significance here? If so, can that be added to the figure?

12. Line 305: Could the authors comment on the possibility that the discrepancy in results between this and previous studies may also be due to the measures used to define T-cell activation – here T-cell numbers and IFNg production (presented as fold-change) were presented. In the study by Grenga et al, activated CD8pos T-cells were defined as “CD107a+, IFNg+ or CD107a+IFNg+”. The raw concentrations of IFNg produced were also presented rather than the fold-change. Brown et al also present ng/mL rather than fold-change. Additionally, comment on the concentration of mAb used in these studies and whether it was different to the study presented here, as this may also contribute to the difference in results.

13. Line 316: Typo “not taken into account” – I think the authors mean “not taking into account”

14. Line 324: Typo “MHI1”, should be “MIH1”.

If these points can be addressed, I would recommend this manuscript for publication.

Author Response

(The authors gave the same response as above.)

Round 2

Reviewer 1 Report

Answer to comment (2):“In the figure legends, “n” refers to the number of independent experiments, i.e. experiments performed at different time points.”It is still unclear how many donors were processed for an experiment at any given timepoint. Pooling data obtained at different timepoints is very tricky as the authors know. So when the revised figure legends now state “n” independent experiments, does that mean n repeat timepoints, with cells from all the 9 donors used per timepoint?

NB: Nowadays, separate datapoints are often overlayed on bar graphs to provide a clearer picture of data replicates.

Figure 3a: for relative MFI, the ratio rather than arithmetic substraction of MFIs should be used, and MFI of the FMO, not the isotype control should be used as the denominator. It is correct that isotype control can be used to ascertain the effectivity of your Fc blocking protocol, but FMO should be used as background signal for further analysis as it allows to determine the extent of spectral spread between channels. There is a lot of not-so-very-recent methodological literature on this.

Figure 3c: huge error bar, no statistical significance, and data presumably not sufficient to claim “strong increase in expression”. I don’t think this graph is of much use. Why not show the PD-1 expression pre/post by flow in figure 3b?

Figure 3f: missing short description of proliferation assay used

Author Response

Concerning: Minor revision of the manuscript entitled “Single domain antibody-mediated blockade of programmed death-ligand 1 on dendritic cells enhances CD8 T-cell activation and cytokine production”, reference 540420.

Dear Editor,

We would like to thank the reviewers for their minor revision remarks. We believe that by taking the reviewers’ comments into account, we were able to strengthen the manuscript. Therefore, we hope that the manuscript in its present form fits all requirements for publication in Vaccines.

Enclosed, we provide our feedback to the reviewers’ questions in a point-by-point reply. In the manuscript, we indicated changes with track changes.

Sincerely,

Katrijn Broos

Reviewer 3 Report

The revised manuscript by Broos et al has been greatly improved by the additional experiments, additions to the text and changes to data presentation. The extra work that has been put into this manuscript has enhanced the clarity of the key messages and increased the impact of the study. I recommend the manuscript for publication after a few minor edits. Comments and minor edits are listed below:

Lines 197-198 – TCRpos – pos should be superscript for consistency. Thank you to the authors for the additional information in this figure clarifying the number of independent experiments, this helps to interpret the data. The additional figure showing comparable levels of TCR expression on PD-1pos and PD-1neg T cells is also helpful.

Figure 2 – the addition of the flow plots really helps to clarify both the methodology and the results. Thank you to the authors for including these.

Line 219-220 “… showing expression of PD-1, admitting at low levels [Figure 2b ; Figure 3b]” – should this read “…showing expression of low levels of PD-1”?

The addition of the avelumab data to figures 1 and 3 is useful. Thank you to the authors for including this.

Figure 3e – this cytokine data is much clearer than how it was previously presented. It would have been useful to know the proportion of T cells producing cytokines (ie by flow) and to assess whether T cells were mono- or polyfunctional. However, the limitations of small percentages of antigen-specific cells (as the authors have notes in their responses) are understandable.

Figure 3f – This graph is slightly unclear. This is the data from the Cell Trace Violet dilution, therefore it may be clearer to present this as percentage of proliferating cells. If I have interpreted the results correctly, then it is also surprising (given the large differences in cell numbers shown in 3d) that the K2 treated cells show just over 2x the proliferation seen in the untreated cells. Can the authors comment on this?

The data are much clearer with just the DC-MEL results presented, rather than the DC-MEL and TriMix DC-MEL. This has helped to clarify the key messages of the study.

Author Response

(The authors gave the same response as above.)
